# Nucleolar Structure and Function in Trypanosomatid Protozoa

**DOI:** 10.3390/cells8050421

**Published:** 2019-05-08

**Authors:** Santiago Martínez-Calvillo, Luis E. Florencio-Martínez, Tomás Nepomuceno-Mejía

**Affiliations:** Unidad de Biomedicina, Facultad de Estudios Superiores Iztacala, Universidad Nacional Autónoma de México, Av. de los Barrios 1, Col. Los Reyes Iztacala, Tlalnepantla CP 54090, Estado de México, Mexico; luisef@unam.mx

**Keywords:** nucleolus, trypanosomatid parasites, Pol I, rRNA, ribosome biogenesis, snoRNA, snoRNP

## Abstract

The nucleolus is the conspicuous nuclear body where ribosomal RNA genes are transcribed by RNA polymerase I, pre-ribosomal RNA is processed, and ribosomal subunits are assembled. Other important functions have been attributed to the nucleolus over the years. Here we review the current knowledge about the structure and function of the nucleolus in the trypanosomatid parasites *Trypanosoma brucei*, *Trypanosoma cruzi* and *Leishmania* ssp., which represent one of the earliest branching lineages among the eukaryotes. These protozoan parasites present a single nucleolus that is preserved throughout the closed nuclear division, and that seems to lack fibrillar centers. Trypanosomatids possess a relatively low number of rRNA genes, which encode rRNA molecules that contain large expansion segments, including several that are trypanosomatid-specific. Notably, the large subunit rRNA (28S-type) is fragmented into two large and four small rRNA species. Hence, compared to other organisms, the rRNA primary transcript requires additional processing steps in trypanosomatids. Accordingly, this group of parasites contains the highest number ever reported of snoRNAs that participate in rRNA processing. The number of modified rRNA nucleotides in trypanosomatids is also higher than in other organisms. Regarding the structure and biogenesis of the ribosomes, recent cryo-electron microscopy analyses have revealed several trypanosomatid-specific features that are discussed here. Additional functions of the nucleolus in trypanosomatids are also reviewed.

## 1. Introduction

The nucleolus, the factory of the ribosomal subunits (r-subunits), is the largest nuclear body within the interphase nucleus of all eukaryotes described to date. At the end of each mitotic phase, this organelle is assembled in the vicinity of the nucleolar organizer region (NOR), a distinctive chromosomal locus described in maize cells by Barbara McClintock more than 80 years ago [1]. The NOR is structured by tandem repeats of the ribosomal RNA (rRNA) genes and a vast number of nucleolar proteins [2]. Competent rRNA gene repeats are recognized and transcribed by DNA-dependent RNA polymerase I (Pol I) and a large number of associated transcription factors [3]. The enzymatic action of this multiprotein complex catalyzes the production of pre-rRNA in the boundary between the fibrillar center (FC) and the dense fibrillar component (DFC) subdomains of the nucleolus (Figure 1A) [4,5]. As part of the co- and post-transcriptional processing, the nascent transcript undergoes a series of exo- and endo-nucleolytic cleavages concurrent with the chemical modification of a high number of sequence-defined nucleotides. These processes are carried out by small nucleolar ribonucleoprotein particles (snoRNPs) which contain small nucleolar RNAs (snoRNAs) that base-pair with complementary rRNA sequences [6]. The vast majority of these cellular events are initiated in DFC and conclude within the granular component (GC), which represents the third subdomain of the tripartite nucleolus (Figure 1A) [7]. In parallel, the individual rRNA species are packaged in the form of r-subunits. The 40S or small subunit (SSU), responsible for deciphering the information encoded in messenger RNAs (mRNAs), comprises a chain of 18S rRNA and 33 different ribosomal proteins (r-proteins). The 60S or large subunit (LSU), which catalyzes the peptide bond formation, is composed of 5.8S and 28S rRNAs, 47 r-proteins, and 5S rRNA (a product of Pol III transcription). Finally, each r-subunit leaves the nucleolus prior to interact with the nuclear pore complexes to be individually translocated to the cytoplasm for the last maturation steps and the assembly of a functional ribosome [8]. The traditional view of the nucleolus as an organelle exclusively designed to coordinate the ribosomal biogenesis has been challenged by findings that show that this nuclear body is a dynamic entity involved in other fundamental biological events including the assembly of the signal recognition particle [9] and the cellular stress response [10].

The structure and function of the nucleolus have been mainly analyzed in vertebrates and yeast. Consequently, the knowledge about this essential nuclear body in early-branched organisms, such as the trypanosomatid protozoan parasites, is scarce. The closely related trypanosomatids *Trypanosoma cruzi*, *Trypanosoma brucei* and *Leishmania* ssp. (Figure 2A) are single-celled flagellated eukaryotes capable of parasitizing humans and causing a collection of neglected tropical diseases that affect various millions of people, primarily in remote and poor regions in developing nations [11]. These pathogenic protozoa have heteroxenous life cycles. To survive in vertebrate hosts and blood-consuming insects, the parasites complete a complicated process of cell differentiation which is finely regulated by differential gene expression [12,13]. The vector-borne parasite *T. cruzi* causes American trypanosomiasis, also known as Chagas disease, an autochthonous illness in 21 countries in Latin America. Usually, the transmission to the human occurs when metacyclic trypomastigotes, released in feces of infected *Reduviidae* insects, enter through mucous membranes or skin wounds and parasitize the host cells. Within the cytoplasm, infective *T. cruzi* is transformed into the amastigote form that propagates by binary fission and differentiates into bloodstream trypomastigotes before the cell host collapse. Then, trypomastigotes enter the blood and lymph vessels and disseminate to other tissues. Circulating parasites may be ingested by insect vectors during a blood meal. Once in the midgut, trypomastigotes change their shape and proliferate actively as epimastigotes (Figure 2A, *T. cruzi*). Finally, epimastigotes attach to the waxy gut cuticle to become infective metacyclic trypomastigotes [14,15]. Leishmaniasis, a group of diseases with different clinical forms that is caused by at least 20 species of the *Leishmania* genus, is endemic in over 98 countries. The initial transmission to the human occurs when highly motile metacyclic promastigotes are inoculated by an infected female sandfly while feeding blood. This infective form is engulfed by mononuclear phagocytes (mainly macrophages), where the parasite transforms to the amastigote stage. This form divides by binary fission within a parasitophorous vacuole and is released into blood after host cell lysis. Amastigotes may infect other macrophages to spread the infection or may be taken up by the sandfly vector (*Phlebotomus* or *Lutzomyia*), where they transform into procyclic promastigotes (Figure 2A, *L. major*), multiply by binary fission, and differentiate to become infective metacyclic promastigotes [16]. Two subspecies of *T. brucei* (*T. b. rhodesiense* and *T. b. gambiense*) are the etiological agents of human African trypanosomiasis, or sleeping sickness, in 36 countries of sub-Saharan Africa. Metacyclic forms of this trypanosomatid are injected in the skin of the host through the bite of the tsetse fly (*Glossina* genus). In the mammal, *T. brucei* replicates as extracellular slender forms in blood, lymph, and cerebrospinal fluid. During the normal course of infection, parasites transform into the quiescent stumpy bloodstream stage that, subsequently, are sucked up by the vector during feeding. The replicative procyclic cells (Figure 2A, *T. brucei*) are generated within the insect midgut before migration towards salivary glands. The life cycle culminates with the appearance of infectious metacyclic trypomastigotes, which are ready to parasitize a new vertebrate host [17,18]. 

In addition to their importance in public health and the global economy, *T. brucei*, *T. cruzi*, and *Leishmania* are relevant in the molecular biology and evolution fields because they exhibit gene expression mechanisms that are exclusive or uncommon within the eukaryotic lineages. Essential cellular events, such as mitochondrial RNA editing [19,20], polycistronic transcription, the maturation of mRNAs by trans-splicing, and the production of some mRNAs by Pol I have been extensively described [21,22,23,24]. This work presents an overview of what is currently known about ribosome structure and biogenesis, as well as the architecture, composition, and putative functions of the nucleolus in trypanosomatids, a group of early-divergent microorganisms. Similarities and differences with the nucleoli of yeast and vertebrates will be highlighted.

## 2. Nucleolar Structure

In several groups of higher eukaryotes (including mammals, birds, reptiles, and plants), the interphase nucleolus is a tripartite nuclear body composed of three major subcompartments that are defined by their morphology, macromolecular content, and function: The FC, the DFC, and the GC (Figure 1A) [7,25]. Ultrastructural analysis of the mammalian nucleoli showed the presence of pale fibrillar regions (the FC) surrounded by electron dense, tightly packed fibrils (the DFC); both are embedded in the GC, the biggest nucleolar subdomain, which is composed of ribonucleoprotein granules of 15–20 nm in diameter [26]. At the beginning of open mitosis, the nucleolus is disintegrated due to the silencing of Pol I activity and the relocation of several nucleolar factors. During telophase, the reorganization of the nucleolus (also called nucleologenesis) is triggered by the reactivation of pre-rRNA synthesis, processing of pre-rRNA, and recruitment of the nucleolar material (agglomerated as prenucleolar bodies) towards the transcriptionally active NOR [27,28,29]. 

As in other eukaryotes, in *Trypanosoma* and *Leishmania* parasites, the nucleolus is the most distinctive membrane-less nuclear body distinguished by light and electron microscopy [30,31]. Throughout interphase, trypanosomatids show a single, spherical, and small nucleolus (Figure 2). At the present time, FCs have not been found in the nucleolus of these parasites, which is observed as a bipartite structure constituted only by a prominent GC that encloses a slight DFC (Figure 1B) [30,31,32]. Similar bipartite nucleoli are found in other organisms, including other protozoan, yeast, invertebrates, fish, and amphibians [7,26,33,34,35]. In contrast with higher eukaryotes, *Trypanosoma* and *Leishmania* undergo a closed mitosis in which the nuclear envelope remains intact, the chromatin does not condense, and the nucleolus persists as an intranuclear, ribonucleoproteic organelle during the whole cell division (Figure 2B) [30,36,37,38]. As mitosis progresses, the nucleolus lengthens and is pulled, via the spindle fibers, towards the opposite poles of the nucleus. Eventually, the nucleolar structure is divided into two large ribonucleoprotein complexes that are transmitted to each nascent nucleus without intervention of intermediary structures, such as the classical prenucleolar bodies (Figure 2B). Finally, late in mitosis, new nucleoli are clearly observed before partition of the cytoplasm by cytokinesis (Figure 2B) [36,37,38]. 

## 3. Ribosomal RNA Genes

Eukaryotic ribosomes are composed of 18S, 5.8S, 28S, and 5S rRNA molecules and around 80 r-proteins. The four rRNA molecules constitute the main structural and catalytic elements of the ribosome. In most organisms, genes encoding 18S, 5.8S, and 28S rRNA are organized as tandem repeats separated by intergenic spacers. The three genes are transcribed together by Pol I, generating a primary transcript (~35–47S) that requires processing to produce the mature 18S, 5.8S, and 28S rRNAs [39,40]. Each rRNA gene repeat contains regulatory sequences that include promoters, enhancers, and terminators, which are located within the intergenic spacer [41]. The precursors of rRNA are synthesized in the boundary between FC and DFC of the nucleoli [4,5]. 5S rRNA genes, on the other hand, are commonly transcribed in the nucleoplasm by Pol III. 

### 3.1. rRNA Gene Repeats

A unique feature of ribosomes in trypanosomatids is the fragmentation of the 28S-type rRNA chain into two large (24Sα and 24Sβ) and four small independent rRNA molecules known as sr1, sr2, sr4, and sr6 in *T. brucei* (γ, δ, ε and ξ in *Leishmania*) (Figure 3) [42,43,44]. Some of the rRNA gene repeats in *L. major* possess two copies of the sr4 (ε) rRNA gene [45]. Interestingly, the cryo-electron microscopy structure of the *T. brucei* ribosome showed that the sr2, sr4, and sr6 molecules interact with one another in the LSU, as well as with a kinetoplastid-specific expansion segment (known as KSD) from the 24Sβ rRNA, to form the missing last domain of the LSU rRNA (that corresponds to domain VI in yeast 28S rRNA) [46]. Notably, it seems that the processing of the LSU rRNA into six independent rRNA species is a structural requirement to create a functional ribosome in trypanosomatids, as the presence and conformation of the KSD in the observed ribosomal structure is conceivable only if the LSU rRNA precursor is fragmented into several pieces [46,47].

Among different eukaryotes, the number of rRNA gene repeats fluctuates from ~100 to more than 10,000; these are usually located on several chromosomes [3,48]. Nevertheless, the *L. major* genome contains only ~12 copies of the rRNA gene repeat per haploid genome, organized in head-to-tail tandem arrays on chromosome 27 [45,49]. Similarly, *T. brucei* possesses only 15–20 rRNA gene repeats that are divided over six to seven chromosomes [50]. Thus, trypanosomatids contain a reduce number of rRNA gene repeats. 

### 3.2. Transcription of the rRNA Gene Repeat

#### 3.2.1. Promoter Regions

Generally, the sequence of the promoter regions of the rRNA gene repeats is not conserved across species. However, most promoters have a common structural organization, as they contain a core element and an upstream control element (UCE). The core element is required for accurate transcription initiation, and the UCE stimulates transcription. The distance and relative orientation of these two elements are critical for promoter activity [3,40]. 

The promoter region of rRNA genes has been characterized in several trypanosomatids. In *T. brucei*, it is constituted by a bipartite core element (domains I and II) and a distal element (domain III) that corresponds to the UCE (Figure 3) [51,52,53,54]. The promoter possesses an additional upstream control region (domain IV), with small influence on transcription efficiency, which extends approximately to position −250 relative to the transcription start site (Figure 3). The rRNA gene promoter is not conserved among trypanosomatids, as *T. cruzi* and different species of *Leishmania* present smaller promoter regions that apparently lack upstream control elements [55,56,57,58,59,60]. Interestingly, Pol I transcriptional repressors were identified upstream of the core promoter region in *L. amazonensis* [61] and downstream of the transcription start site in *T. cruzi* [62]. Within the intergenic regions, there are repeated sequences of 60 to 64 bp in *Leishmania* spp. [45] and 172 bp in *T. cruzi* [63] that seem to regulate Pol I transcription. 

Among trypanosomatids, transcription termination in rRNA genes has been mainly analyzed in *Leishmania*. An early study in *L. infantum* showed that transcription ends in an area that contains short sequences with the potential to form stem-loop structures located downstream of the 3′ end of the rRNA gene repeat [64]. Similar sequences, which are reminiscent of the bacterial rho-independent transcriptional terminators, were observed in the *L. major* rRNA genes, in a region where run-on experiments revealed that transcription ceases [45]. The presence of two transcriptional terminators in this region (named T1 and T2) was confirmed in a functional analysis performed in *L. amazonensis* [65]. This report showed that T1 (located 185 bp downstream of the sr4 rRNA gene) is the main Pol I transcription terminator, whereas T2 (found 576 bp downstream of sr4 rRNA gene) acts as a backup terminator. A CCCTTTT motif, present in both T1 and T2, is needed for ending transcription [65].

#### 3.2.2. Transcription Factors and Pol I Subunits

In vertebrates, the recruitment of Pol I to the rRNA gene promoter, for the assembly of the pre-initiation complex (PIC), is principally directed by three general transcription factors: Selectivity factor 1 (SL1), RRN3, and the upstream binding factor (UBF) [3,39]. SL1 consists of the TATA binding protein (TBP) and several TBP-associated factors (TAFs), including TAFI110, TAFI63, TAFI48, and TAFI41. SL1 recognizes and binds the core promoter element to recruit Pol I by interacting with the Pol I-associated factor RRN3. To activate Pol I transcription, UBF is incorporated into the PIC by making contact with SL1 and the UCE domain of the promoter [40]. 

Orthologues of SL1, UBF, and RRN3 have not been identified in trypanosomatids. Though they contain a TBP-related factor (TRF4), it has been shown that it binds to the rRNA coding regions but, interestingly, not to the promoter sequences [66]. Nonetheless, an essential factor for Pol I transcription, class I transcription factor A (CITFA), was purified and characterized in *T. brucei* [67]. CITFA consists of seven trypanosomatid-specific proteins, which were called CITFA-1 to -7, and the dynein light chain LC8 [68]. In *T. brucei*, CITFA binds to the rRNA gene promoter, as well as the promoter regions from the variant surface glycoproteins (VSG) and procyclins [69], which are also transcribed by Pol I in this parasite (see below). 

Another protein that regulates rRNA gene transcription in *T. brucei* is Elp3b. While Elp3 orthologues in other species are part of the Elongator complex that controls Pol II transcription elongation, in *T. brucei*, Elp3b is a nucleolar protein that negatively regulates Pol I transcription of the rRNA genes [70]. In *L. major*, Elp3b is also a nucleolar protein [71], which suggests that its role in the control of rRNA gene transcription is conserved across trypanosomatids. These parasites possess a second isoform of Elp3, called Elp3a, whose function is unknown [70]. 

Yeast Pol I consists of 14 subunits, and there are mammalian orthologues for all but subunit RPA14 [40]. In *T. brucei*, in silico analyses and tandem affinity purifications have led to the identification of ten Pol I subunits: RPA1, RPA2, RPA12, RPC19, RPC40, RPB5z, RPB6z, RPB8, RPB10z, and RPB12 [72,73,74]. Orthologues of the two heterodimers formed by subunits RPA14/RPA43 and RPA49/RPA34.5 have not been identified in trypanosomatids. Subunits RPB5z, RPB6z, and RPB10z, which are only present in trypanosomatids, are paralogues of subunits RPB5, RPB6, and RPB10 that are exclusive to Pol I [54,75]. Moreover, a novel trypanosomatid-specific Pol I subunit, named RPA31, was identified in *T. brucei* [74]. The silencing of RPA31 by RNA interference is lethal, as it affects rRNA abundance [74]. In growing *T. cruzi* cells, RPA31 localizes to the nucleolus, but, in nonproliferative cells, it delocalizes to the nucleoplasm [76]. 

#### 3.2.3. Epigenetic Regulation of rRNA Gene Repeats

In most species, only a fraction of rRNA gene repeats is transcribed at any given time [77], and it has been established that chromatin structure plays an important role in the silencing and activation of these genes [78,79,80]. While inactive rRNA gene repeats show a tightly packaged chromatin structure characterized by repressive histone modifications such as trimethylation of lysine 9 in histone H3 (H3K9me3) [81], transcriptionally active rRNA gene repeats present an open chromatin state distinguished by acetylated histones [40,77,82]. 

In *L. major*, the promoter region of the rRNA gene repeat is practically devoid of nucleosomes, whereas the intergenic spacer presents a tight nucleosomal structure. Intermediate levels of nucleosomes were observed in the rRNA coding regions [83]. A similar distribution of nucleosomes is present in rRNA genes in *T. brucei* [84,85]. It is worth noting that, unlike other organisms, all the ∼12 copies of the rRNA gene repeat in *L. major* could be transcriptionally active in exponentially growing cells, as the nucleosomal patterns observed in the rRNA genes in this parasite strongly resemble those found in a yeast mutant that possesses only active rRNA genes [83,86].

Notably, chromatin immunoprecitation (ChIP) experiments showed that the *L. major* rRNA promoter region contains several histone modifications that are usually associated with activation of transcription, including H3K14ac, H3K23ac, and H3K27ac [83,87]. In *T. brucei*, the subunits of the chromatin-remodeling ISWI complex are enriched in the intergenic spacer of the rRNA gene repeat, indicating that this complex might regulate transcription of rRNA genes [88]. Moreover, subunit Spt16 of the histone-chaperone FACT complex seems to regulate the processivity of Pol I transcription in the *T. brucei* rRNA genes [89]. Interestingly, the knockdown of the high mobility group protein TDP1 produced a decrease in rRNA precursor transcripts in *T. brucei*, showing that it is required for synthesis of rRNA in this parasite [90]. Thus, epigenetic mechanisms play key roles in the regulation of rRNA gene transcription in trypanosomatids [87]. 

### 3.3. Gene Organization and Transcription of 5S rRNA Genes

Unlike the rRNA gene repeat transcribed by Pol I, 5S rRNA genes are transcribed by Pol III [91]. In several yeast species, 5S rRNA genes are attached to the rRNA transcription unit, but they are oriented in the opposite direction [92]. However, in most organisms 5S rRNA genes are found in tandem head-to-tail repeats in one or several loci [41]. In trypanosomatids, 5S rRNA genes are organized into tandem arrays in *T. brucei* and *T. cruzi* [93,94], but they are dispersed throughout the genome and are linked to tRNA genes in the different species of *Leishmania* [49,71]. While several hundreds of 5S rRNA genes are estimated to be present in *T. brucei* and *T. cruzi* [93,94], *Leishmania* species contain only from nine to eleven copies of the 5S rRNA gene [71]. Therefore, the genomic organization and number of 5S rRNA genes differ substantially between *Leishmania* and *Trypanosoma*. 

In yeast species, 5S rRNA genes are located within the nucleolus, as they are linked to the rRNA gene repeat. However, 5S rRNA genes may be located in the nucleolar vicinity in several species where these genes are not attached to the rRNA gene repeat in the linear DNA [95]. For example, a clear association of one 5S rRNA gene array and the nucleolar periphery was reported in the plant *Pisum sativum* [96], and human 5S rRNA genes are regularly found at the nucleolar margin [97]. In contrast, the 11 copies of the 5S rRNA gene present in *L. major* are predominantly located at the nuclear periphery [71]. In *T. brucei* 5S rRNA genes are mainly located in a central position within the nucleus [98], and in *T. cruzi* they are distributed throughout the nucleus [71].

The transcription of 5S rRNA genes by Pol III is directed by a type 1 promoter, which consists of three elements located within the transcribed region: Box A, the intermediate element (IE), and box C [99,100]. In trypanosomatids, 5S rRNA genes contain these three internal elements [93,94,101], but they have not been functionally characterized. Box A is very similar between *L. major* and *S. cerevisiae*, since nine out of 15 nucleotides are conserved. Box C is the least conserved promoter element, as only seven out of 18 bases are conserved between *L. major* and *X. laevis* [71]. The majority of 5S rRNA genes in the different species of *Leishmania* are syntenic, and their sequence is highly conserved. Notably, the most variable nucleotide lies within the IE, suggesting that the differential expression of 5S rRNA genes takes place in *Leishmania* [71].

Regarding transcription factors that regulate 5S rRNA transcription in trypanosomatids, ChIP-chip analysis in *L. major* showed that TRF4 and SNAP50 bind to all tRNA, snRNA and 5S rRNA gene clusters [66]. In addition, ablation of subunit Brf1 of TFIIIB in *T. brucei* reduced the expression of all Pol III-dependent genes analyzed, including 5S rRNA genes [102]. Interestingly, contrary to what occurs in other organisms, the downregulation or overexpression of the negative regulator Maf1 did not significantly change the levels of 5S rRNA in *T. brucei* [103]. 

In trypanosomatids, little is known about chromatin architecture of 5S rRNA genes. In *L. major*, Southern blots with nucleosomal ladders revealed that 5S rRNA and tRNA genes possess an open chromatin structure, as they show a marked smearing in the micrococcal nuclease profile [83]. In accord with this finding, a genome-wide analysis in *L. major* demonstrated that clusters of genes transcribed by Pol III are nucleosome-free regions [104]. This is different from protein-coding genes, which present a strong and regularly-spaced nucleosomal structure that might reflect a relatively low transcriptional rate of Pol II in *L. major* [83,87,104].

## 4. Processing and Nucleotide Modifications in rRNA

The transcription of each rRNA gene repeat produces a long primary transcript that requires processing to generate mature 18S, 5.8S and 28S rRNAs. Notably, while rRNA processing initiates in the nucleolus, the final maturation steps take place in the cytoplasm, after export of the pre-40S and pre-60S ribosomal subunits [105]. 5′ and 3′ external transcribed spacers (ETSs) and internal transcribed spacers (ITSs) are removed from the primary transcript by snoRNAs and multiple proteins that include endo- and exo-nucleases, RNA helicases, RNA chaperones, ATPases and GTPases [106]. All rRNAs species contain pseudouridine residues and 2′-O-methyl groups that are required for proper function of the ribosome.

### 4.1. Roles of snoRNAs in rRNA Maturation

Modifications in rRNA are directed by snoRNAs, which are classified into two types, box C/D and box H/ACA. While box C/D snoRNAs guide methylation, box H/ACA snoRNAs participate in pseudouridylation [99,107]. Nucleotide selection for modification occurs by transient base-pairing between snoRNA and rRNA. Some snoRNAs from both types are required for rRNA processing. Several proteins associate with snoRNAs to form snoRNPs. The box C/D snoRNPs contain four proteins: The methyltransferase fibrillarin (also known as Nop1), Nop56, Nop58 and Snu13. The box H/ACA snoRNPs are also comprised of four proteins: The Cbf5 pseudouridine synthase, Nop10, Gar1 and Nhp2 [99,107].

Regarding box C/D snoRNPs, trypanosomatids possess orthologues of all four proteins. In *T. cruzi*, fibrillarin contains the two typical regions: The methyltransferase catalytic domain and the N-terminal glycine- and arginine-rich (GAR) domain [108]. Knockdown of fibrillarin by RNAi showed that it is essential for growth of procyclic forms of *T. brucei*. Fibrillarin-silenced cells showed defects in methylation and processing of the 18S rRNA [109]. Similarly, ablation of Nop58 demonstrated that it is essential for *T. brucei* viability [109]. The *T. brucei* Snu13 protein shares 64% identity with the yeast orthologue and, as expected, it interacts with snoRNAs [109]. In *L. major*, in silico analyses showed that, despite sequence divergence, Nop56 contains the three structural and evolutionary conserved domains (NOP5NT, NOSIC and Nop) and that its predicted three-dimensional structure is very similar to that of the yeast orthologue [38].

Trypanosomatids also possess orthologues of the Cbf5 pseudouridine synthase, Nhp2, and Nop10, integral components of the H/ACA snoRNPs [110,111]. In *T. brucei*, Cbf5 is essential for viability, and its knockdown generated severe defects in rRNA processing. Though the *T. brucei* Nhp2 protein shares only 33% identity with the yeast orthologue, it was demonstrated that it associates with box H/ACA snoRNAs [110]. Nop10 has not been functionally characterized in trypanosomatids, but it was shown that it localizes in a central nucleolar position in the *T. brucei* procyclic form [111]. A putative orthologue of Gar1, the forth component of box H/ACA snoRNPs, is annotated in the trypanosomatid databases. 

The genomic organization of snoRNAs differs across eukaryotes. Whereas most snoRNA genes in yeast are independent, the majority of vertebrate snoRNAs are encoded in introns of host genes [112,113]. In trypanosomatids, most snoRNA genes are organized in clusters that are transcribed polycistronically by Pol II. The *T. brucei* genome contains 142 snoRNA genes (79 box C/D and 63 box H/ACA) [114]. Similarly, the *L. major* genome encodes for 161 snoRNAs (80 box C/D and 81 box H/ACA) [115]. In these parasites, all snoRNAs, either independent or clustered, are processed from precursor molecules that are trans-spliced and polyadenylated [116]. 

In most organisms, a small subset of around six snoRNAs participates in rRNA processing, including the box C/D snoRNAs U3, U8, U14 and U22, and the box H/ACA snoRNAs U17 (snR30) and snR10 [92]. Remarkably, at least 18 snoRNAs (16 box C/D and 2 box H/ACA) are involved in rRNA processing in trypanosomatids, which is the highest number ever reported [117]. This is in accordance with the extensive processing required to produce the six independent rRNA species from the LSU precursor in trypanosomatids (Figure 3). As anticipated, several of these snoRNAs are exclusive to trypanosomatids [117]. In yeast, U3 snoRNA is required for early processing events conducting to 18S rRNA maturation, as it participates in cleavage at A0, A1 and A2 sites [106]. In *T. brucei*, U3 snoRNA base-pair with two regions of the 5′-ETS, and this interaction is needed for processing the 18S rRNA precursor at sites A0 and A1 (Figure 3) [118,119]. Similarly to other organisms, box H/ACA snR30 (U17) and MRP snoRNA participate in early processing of the rRNA precursor in *T. brucei* [109,110]. Other snoRNAs involved in rRNA processing in *T. brucei* are: TB11Cs2C1 (U31), TB11Cs2C2, TB10Cs4C4, TB6Cs1C3, TB9Cs2C1, TB9Cs2C5 (snR60), TB9Cs3C3, TB10Cs4C3, TB11Cs3C2, TB10Cs1C4 (snR75), TB10Cs1C1 (snR64), TB8Cs1C1, TB8Cs1C3, TB9Cs2C3 (snR39b) and TB9Cs3H2 (Atsnor77, a box H/ACA snoRNA) [117,120,121]. Some of these snoRNAs are not directly involved in cleavage, but guide modifications required for rRNA processing.

### 4.2. Primary Transcript Processing

In addition to the 5′ and 3′-ETS, the rRNA primary transcript in trypanosomatids contains seven ITSs that separate the eight mature rRNA species (Figure 3). Processing of the rRNA precursor has been analyzed in *T. brucei* by several groups [118,121,122,123,124]. While in other eukaryotes the rRNA precursor is first cleaved at the 5′-ETS, processing of the ~9.6 kb rRNA precursor in *T. brucei* starts with a cleavage within ITS1 (at site B1) to separate the SSU rRNA (18S) (3.7 kb precursor) from the LSU rRNAs (5.8S, 24Sα, 24Sβ, sr1, sr2, sr4, and sr6) (5.9 kb precursor) (Figure 3). Then, the 5′-ETS is eliminated from the 3.7 kb molecule by successive cleavage at sites A′, A0, and A1 (producing intermediates of 3.6, 2.6 and 2.5 kb). The 2.5 kb intermediate is then cleaved at the A2 site (at/near the 18S/ITS1 boundary) to produce the mature 18S rRNA. Simultaneously, the 5.8S rRNA is separated from the LSU precursor (5.9 kb molecule) via cleavage within ITS2, generating the 5.1 and 0.6 kb intermediates. Next, a cleavage within ITS5 separates the two large rRNAs (24Sα and 24Sβ) and sr1 from the three distal small rRNAs (sr2, sr4, and sr6) (Figure 3) [119,125,126]. Though the remaining steps to produce the mature rRNA molecules have yet to be established, recent data indicate that sr1 is the last LSU rRNA to be processed [117,127].

In trypanosomatids, only a small number of proteins that participate in rRNA processing has been identified in *T. brucei*. These include NOG1, whose disruption by RNAi led to the generation of an unusual rRNA intermediate in which ITS2 was not cleaved [124]. NOP44/56, a trypanosomatid-specific nucleolar phosphoprotein that associates with NOG1, also participates in LSU rRNA processing [125]. It was also established that pumilio domain protein PUF7 is needed for effective cleavage of the 9.6 kb precursor, and it seems to also participate in processing of the 3.7 kb precursor [126]. Another pumilio domain protein, PUF10, as well as its interaction partners NRG1 (Nucleolar Regulator of GPEET 1) and BOP1, participate in maturation of 5.8S rRNA in procyclic forms of *T. brucei* [128]. Moreover, the 5′→3′ exoribonuclease XRNE participates in processing of *T. brucei* rRNA, as its depletion by RNAi produces the accumulation of aberrant 18S and 5.8S rRNAs. The XRNE is also involved in the biogenesis of the LSU and assembly of polysomes [129]. In addition, the ablation of protein uL5 showed its participation in the maturation of precursor rRNAs [130]. Moreover, it was recently shown that PNO1 (a KH-domain protein) and ribonuclease NOB1 participate in cleavage activity at sites A1, A2, and B1 [131]. Another recent work demonstrated that ablation of the *T. brucei* UTP10 homologue (also known as Bap28 and HEATR1) disrupts 18S rRNA processing [132]. 

### 4.3. Modifications in rRNA Molecules

The most abundant modified nucleotide in rRNA is pseudouridine, which may be involved in rRNA folding and ribosome assembly. In the mature ribosome, pseudouridines might help to stabilize local secondary and tertiary structures through RNA-RNA and RNA-protein interactions [133]. The Cbf5 pseudouridine synthase present in H/ACA snoRNPs is responsible for the isomerization of uracil to pseudouridine [134]. In trypanosomatids, an in silico analysis predicted the presence of a large number of pseudouridines in the rRNA molecules [115]. A subsequent genome-wide mapping by PSI-Seq confirmed the existence of 68 pseudouridine sites in the *T. brucei* rRNA. Notably, 31 of these pseudouridine sites are trypanosome-specific [135]. Of the 68 pseudouridine sites found in the rRNA, 21 are hypermodified in the bloodstream form of the parasite. Some of the hypermodified positions are located in the peptidyl-transferase center, where they probably support the function of the ribosome in this stage of *T. brucei* [135]. Interestingly, the overexpression of the snoRNAs that guide pseudourydylation in these hypermodified positions enhanced the growth of procyclic forms at high temperatures, which suggests that hyper- pseudouridylation may help the parasite to adapt during the transition from the insect to the vertebrate host [135]. 

Ribose 2′-O-methylation in rRNA is essential for efficient protein synthesis in eukaryotic cells. In humans, rRNA methylation is modulated in ribosomes, and changes in the pattern of methylation regulate the abilities of ribosomes to translate mRNAs [136]. The methyltransferase fibrillarin, contained in box C/D snoRNPs, catalyzes 2′-O-methylation of rRNA. Mapping by primer extension allowed the identification of 131 methylated residues in the *T. brucei* rRNA [109]. Interestingly, 60% of these modifications are specific to *T. brucei*, and they are localized outside the known functional ribosomal domains. It was also found that several nucleotides located in important functional regions of rRNA are hypermethylated in the bloodstream form of the parasite, which suggests that methylation may help the parasite to deal with the temperature changes that occur when cycling between the two hosts [109]. 

The presence of a larger number of modified rRNA nucleotides in trypanosomatids, compared to other organisms, was confirmed by direct visualization of the *L. donovani* ribosome with high-resolution cryo-electron microscopy [127]. Similarly to prokaryotic organisms, the majority of the modified nucleotides were found in internal regions of the ribosome, and not at the outside edge. It was suggested that modifications located in the vicinity of fragmented rRNA terminals may enhance the stability of the segmented LSU rRNA that these parasites possess [127]. 

## 5. Ribosome Biogenesis

The translation of mRNAs into proteins is an essential activity carried out by the ribosomes in all the kingdoms of life. The formation of both r-subunits is intimately associated to the status of growth and proliferation of cells. In yeast, ribosome biogenesis is a vectorial multi-stage process that begins in the nucleolus, with the transcription and processing of the pre-rRNA, as well as the incorporation of the 5S rRNA into pre-60S particles. The process continues with some maturation steps in the nucleoplasm and concludes in the cytoplasm with the cleavage of both 20S and 6S pre-rRNA to produce the mature rRNA species 18S and 5.8S, respectively [8,137,138,139]. The correct building and function of r-subunits need the transient association of a large number of trans-acting factors such as snoRNAs and more than 200 different non-ribosomal proteins, including RNA-binding proteins, endo- and exo-nucleases, methyltransferase, pseudouridine synthase, RNA helicases, GTPases, and ATPases [8,138,139,140], as well as the sequential incorporation of each r-protein [138]. The knowledge of the architecture of the ribosomes and the molecular mechanisms involved in their biogenesis was increased with cryo-electron microscopy analyses, which made it possible to distinguish rRNA nucleotides, chains of amino acids, and modifications of rRNA species of the individual r-subunits [141,142], as well as the complete 80S ribosomes from *S. cerevisiae* [143] and humans [144]. Ribosomal biogenesis in *S. cerevisiae* has been meticulously analyzed and often serves as a comparative prototype for other eukaryotic cells.

In trypanosomatids, the efficient translation of mRNAs containing a hypermodified 5′ cap motif (the most modified cap known in eukaryotic cells) is carried out through unusual cytoplasmic ribosomes, which have a LSU formed by eight distinct modules of rRNA (5.8S, 5S, 24Sα, 24Sβ, sr1, sr2, sr4 and sr6) (Figure 3). As one might suppose, the multi-fragmentation of LSU rRNA significantly increase the complexity of ribosome biogenesis. Cryo-electron microscopy analyses indicate that the general three-dimensional architecture of the ribosomes from *Trypanosoma* [46,145] and *Leishmania* [146] is similar to that of yeast ribosomes. Nevertheless, at the atomic level there are some distinctive structural features in both r-subunits of trypanosomatids. For instance, the SSU and LSU of these parasites are larger than their yeast counterparts due to the presence of longer expansion segments (ES) in the rRNA, the occurrence of several trypanosomatid-specific ESs (including the KSD), as well as the association of r-proteins that contain large extensions of amino acids, mainly at the N- or C-terminal ends, that are absent in the *S. cerevisiae* orthologues [46,127,145].

As in other organisms, in trypanosomatids, the biogenesis of the r-subunits is a process that starts in the nucleolus and ends in the cytoplasm. Initially, the rRNA primary transcript associates with multiple assembly factors and snoRNAs in the nucleolus to form the 90S pre-ribosome [147]. Then, a cleavage at site B1 (within ITS1) separates the emerging small subunit from the incipient large subunit, both of which go through independent maturation routes (Figure 3). A key player in the initial steps of the LSU assembly is the 5.8S rRNA, which interacts with the 5′ end of the 24Sα rRNA and mediates the association between the 3′ end of 24Sα and the 5′ end of 24Sβ rRNAs to generate the LSU rRNA scaffold. At this point, several r-proteins are already bound to the scaffold. One of them, uL3, helps to recruit the sr2 rRNA, which is also anchored to the scaffold by several contact sites with both 24Sα and 24Sβ. At the same time, the sr6 rRNA is positioned by the eL33 r-protein and by the ES7L of the 24Sα rRNA. Notably, sr6 is smaller than its counterpart in yeast, ES39L, and the space resulting from the absence of two helices is occupied by two trypanosomatid-specific ESs (ES42L and KSD) and by the terminal extensions of the eL14 and eL33 r-proteins. Next, the sr1 species is anchored to the scaffold by contacting r-proteins eL19 and eL34 and by base-pairing with sr2. The sr4 molecule is also positioned by interactions with sr2, as well as with r-proteins uL3 and eL31, and the KSD stretch [46,47,127]. 

Very early in the biogenesis process, the 5S rRNA is imported to the nucleolus and incorporated into the 90S pre-ribosome as a ribonucleoprotein complex that contains r-proteins uL5 and uL11, as well as TbRpf2 and the trypanosome-specific proteins P34 and P37 [147,148,149]. However, it is at late stages of the LSU assembly that the 5S rRNA undergoes the spatial rearrangements that place it in its final position at the central protuberance, which facilitates subsequent maturation steps of the 60S subunit [47]. The pre-60S particle is efficiently translocated from the nucleus to the cytoplasm through protein-protein interactions that P34 and P37 established with the transport factors exportin 1 and Nmd3, and with r-proteins uL3 and uL11 [147]. The proposed r-protein content of LSU of *Trypanosoma* and *Leishmania* includes uL1, uL2, uL3, uL4, uL5, uL6, eL6, eL8, uL11, uL13, eL13, uL14, eL14, uL15, eL15, uL16, uL18, eL18, eL19, eL20, eL21, uL22, eL22, uL23, uL24, eL24, eL27, eL28, uL29, eL29, uL30, eL30, eL31, eL32, eL33, eL34, eL36, eL37, eL38, eL39, eL40, eL42, eL43, and eL44 [47,127,146,150,151]. 

Biogenesis of the 40S r-subunit in trypanosomatids occurs, in general, as described in yeast [152]. The assembly of the SSU requires the stable interaction between the long chain of 18S rRNA (the largest known so far) and a set of r-proteins that includes eS1, uS2, uS3, uS4, eS4, uS5, eS6, eS7, eS8, uS7, uS8, uS9, uS10, eS10, uS11, uS12, uS13, uS14, uS15, uS17, eS17, uS19, eS19, eS21, eS24, eS26, eS27, eS28, eS30, and RACK1 [146,150,151]. Notably, the 40S r-subunit contains a trypanosomatid-specific large helical structure known as the “turret”, formed with segments of both ES6 and ES7, that is located near the mRNA exit channel. It was proposed that the turret might participate in translation initiation by interacting with the spliced leader sequence and its unusually modified cap at the 5′ end of the mature mRNA [46,145].

## 6. Other Functions of the Nucleolus

The proteomic analysis of the human nucleolus revealed the presence of more than 4500 distinct proteins that have been grouped in several categories according to the cellular function in which they participate [153]. The fact that the largest population of the nucleolar proteins is not related to ribosomal biogenesis [153,154] reinforced the multitasking essence of the human nucleolus. Indeed, the available evidence has shown that several processes take place in the nucleolus, including the assembly of the signal recognition particle [9], regulation of cell cycle progression [155], early processing of transfer RNA [156,157], and stress response [10]. Moreover, current evidence indicates that this organelle plays an important role in the progression of some ribosomopathies, degenerative diseases, and cancer [155,158]. 

### 6.1. Stress Response

The mammalian nucleolus is considered the most important cellular stress sensor [159,160]. When cells are exposed to a variety of stressors (e.g., ultraviolet irradiation, heat shock, hypoxia, nutrient starvation, inhibition of rRNA transcription, or aberrant r-proteins) complex signaling pathways are triggered. These intracellular activities converge on the impairment of r-subunits biogenesis (known as nucleolar stress), producing disruption of the nucleolus integrity and release of nucleolar proteins to the nucleoplasm, which can lead to p53 activation and stabilization [159,160,161,162,163]. Damaged cells can be arrested during the cell cycle and activate repair mechanisms or initiate p53-mediated cell death by apoptosis [159,160,161,162,163,164]. In addition, some nucleolar proteins are able to directly promote cell cycle arrest or apoptosis without p53 participation, including the r-proteins uL5 and uL11, NuMa, nucleophosmin, and p19^ARF^ [164].

During the normal course of their heteroxenous life cycles, trypanosomatids display several development stages in response to different stressors present in each host. As mentioned above, morphological changes are accompanied by strict readjustments of gene expression. Under situations of nutritional starvation, such as metacyclogenesis, there is a pronounced decrease in transcriptional activity, mainly of Pol I, causing the dispersion of nucleolar material into the nucleoplasm and the disassembly of the bipartite nucleolus [165], suggesting that the *T. cruzi* nucleolus can act as a true stress sensor and a coordinator of cell response during the differentiation process. This hypothesis is reinforced by the fact that during the stationary phase of culture (which is a suitable in vitro condition that recreates the nutrient stress faced by trypanosomatids during the initiation of metacyclogenesis in vivo), the low rates of RNA synthesis are associated with both the subcellular relocation of several nucleolar antigens and the dramatic alterations of nucleolar ultrastructure [32,38,76,108,166]. In addition, the treatment of *T. cruzi* and *L. mexicana* with actinomycin D, a transcription inhibitor, provoke the nucleolar agglomeration of a select group of RNA binding proteins (RBPs) involved in mRNA metabolism [167,168]. Similar data were obtained when *T. cruzi* parasites were incubated with the antimalarial drug chloroquine, another transcription inhibitor [167]. Furthermore, in epimastigotes of *T. cruzi* a set of cytoplasmic mRNAs is translocated into the nucleolus in response to cellular stress generated by severe heat shock exposure [169]. Moreover, while long-term incubation with actinomycin D results in cytoplasmic mRNA decay, it also causes nucleolar mRNA accumulation, which suggests a potential role of the nucleolus in the protection of mRNAs under stress conditions [167]. Thus, as in other eukaryotes, the trypanosomatid nucleolus seems to coordinate different cellular responses to confront the plethora of cellular stressors, indicating that this bipartite nuclear body participates in non-ribosomal functions. 

### 6.2. Transcription and Processing of mRNAs Encoding Procyclins

In *T. brucei*, unlike other organisms, Pol I is able to transcribe protein-coding genes, in addition to rRNA genes [170]. In bloodstream forms, Pol I synthesizes the mRNAs of the variant surface glycoproteins (VSG), which participate in the process of antigenic variation [171,172]. The expressed VSG gene is found at the 3′ end of a ~50 kb polycistronic unit that includes several genes known as the expression-site-associated-genes (ESAGs) [173,174]. Interestingly, ESAG8 accumulates in the nucleolus and may be involved in cell cycle regulation by interacting with a pumilio protein [175]. The active VSG polycistronic unit is transcribed outside the nucleolus in a small compartment called the expression site body [176]. 

When bloodstream forms differentiate into procyclic forms, the VSG coat is replaced by a new surface coat composed of EP/GPEET procyclins, whose mRNAs are also synthesized by Pol I [170]. The procyclin genes are part of polycistronic transcription units found on chromosomes VI and X. Each polycistronic unit contains two procyclin genes followed by several procyclin-associated genes (PAGs) [177]. Unlike VSGs, the transcription of EP/GPEET procyclin genes takes place at the periphery of the nucleolus [178]. Notably, the Pol I promoters in the procyclin polycistronic units are very similar to the rRNA gene promoter [51,179,180]. Similarly to rRNA and VSG genes, CITFA is required for the transcription of EP/GPEET procyclins [67]. In the nucleolus, mature procyclin mRNAs are generated by trans-splicing and polyadenylation, just like mRNAs produced by Pol II are processed in the nucleoplasm. Interestingly, four proteins involved in rRNA processing (PUF7, PUF10, NRG1, and BOP1), also bind the GPEET procyclin mRNA and act as negative regulators [128]. In addition, an MCM-Binding Protein (TbMCM-BP) and a nucleoplasmin-like protein (NLP) are required for silencing of procyclin and PAGs in the bloodstream forms of *T. brucei* [181,182]. The chromatin remodeller ISWI regulates transcription of the EP procyclin locus in both stages of *T. brucei* [183].

## 7. Conclusions

Trypanosomatids possess a single nucleolus primarily constituted by a granular component that surrounds a slight dense fibrillar component. During mitosis, the nucleolus persists and appears to separate out in a relatively intact form. These parasites contain a large repertoire of snoRNAs that participates in the extensive processing that undergoes the rRNA primary transcript in order to produce eight independent rRNAs. Though their general architecture is conserved, ribosomes in trypanosomatids are larger than in other species due to the occurrence of longer ESs in the rRNA, the existence of some trypanosomatid-specific ESs, and the presence of large extensions in r-proteins. In addition to participate in ribosome biogenesis, the nucleolus in trypanosomatids works as a stress sensor and seems to participate in the regulation of cell response during stage differentiation. In addition, the transcription and processing of procyclin mRNAs takes place in the *T. brucei* nucleolus. Hence, nucleolar plurifunctionality appears to be a feature obtained early in the evolution of the eukaryotic lineages.

## 8. Future Outlook

Recent research has advanced our understanding of the structure and function of the nucleolus in trypanosomatids. Nevertheless, several subjects still remain unexplored in these parasites. For instance, little is known about the protein content and the genomic architecture of NORs. Is CITFA the only transcription factor required for Pol I transcription of rRNA genes? How is a single nucleolus formed around rRNA genes that are distributed in at least six chromosomes in *T. brucei*? The fact that the nucleolus remains intact throughout mitosis suggests that Pol I transcription is not totally repressed during cell division, but this issue has not been addressed. Regarding processing of the rRNA primary transcript, the final steps leading to the generation of the mature sr1, sr2, sr4, and sr6 species have not been determined. Moreover, the identity of the nucleases involved in pre-rRNA cleavage is currently unknown. In addition, the specific function of methylated nucleotides and pseudouridine residues in rRNA processing and ribosome function has yet to be investigated. Proteomic and transcriptomic analyses would help to further explore the multitasking nature of the nucleolus in trypanosomatids. Moreover, these studies would provide critical information for the development of highly selective anti-trypanosomatid drugs. 

## Figures and Tables

**Figure 1 cells-08-00421-f001:**
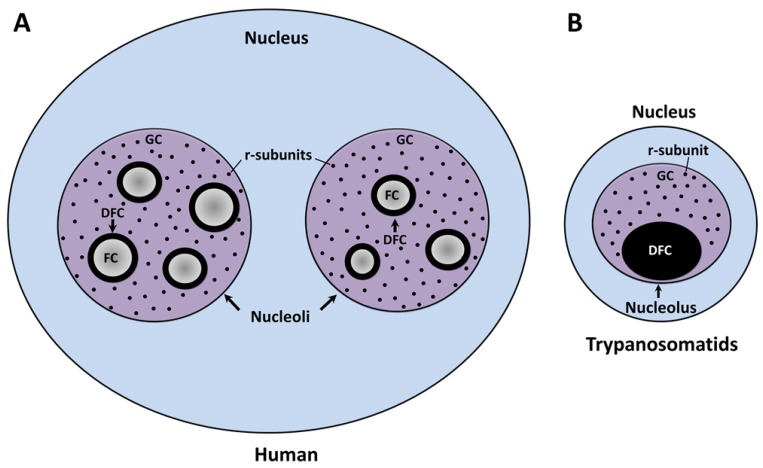
Schematic representation of the ultrastructural architecture of the nucleoli in humans (**A**) and trypanosomatids (**B**). The nucleus in trypanosomatids contains a bipartite nucleolus built by a granular component (GC) and a dense fibrillar component (DFC). The fibrillar centers (FC) are not detected by transmission electron microscopy. Human cells possess tripartite nucleoli that contain FCs, DFCs and GC. The precursors of the r-subunits are represented by black spheres loosely distributed into the GCs.

**Figure 2 cells-08-00421-f002:**
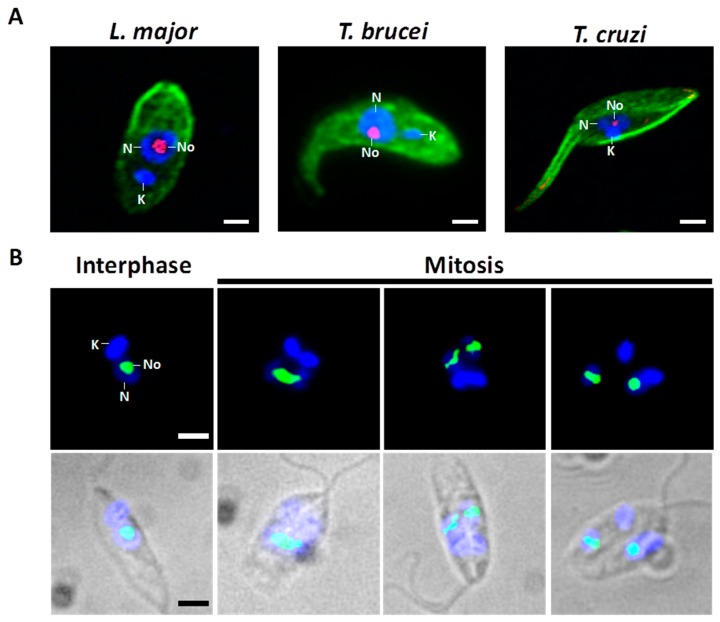
Interphase and mitotic nucleolus in trypanosomatids. (**A**) Fluorescence micrographs of *L. major* (procyclic promastigote stage), *T. brucei* (procyclic form), and *T. cruzi* (epimastigote stage). These three stages, which possess a single flagellum, grow and replicate in the corresponding insect host. They can be grown in large numbers in axenic culture media. These parasites have a single mitochondrion, which contains a network of thousands of catenated circular DNAs known as kinetoplast DNA. Parasites were fixed and treated with antibodies against nucleolar protein Nop56 from *L. major* (red) and *α*/*β*-tubulin (green) for visualization of the nucleolus and microtubules, respectively. Nuclear and kinetoplast DNA were counterstained with DAPI (blue). During interphase, the nucleolus is present as a single structure (red) located in a nucleoplasmic region weakly stained with DAPI. (**B**) Fluorescence images of *L. major* procyclic promastigotes during the cell cycle. Throughout the closed mitosis the nucleolus, represented here by Nop56, is conserved (green signal). During the course of the nuclear division, the round-shaped nucleolus is elongated and, eventually, split into two structures. Nuclear and kinetoplast DNA were counterstained with DAPI (blue). K, kinetoplast; N, nucleus; No, Nucleolus. Bar, 2 μm.

**Figure 3 cells-08-00421-f003:**
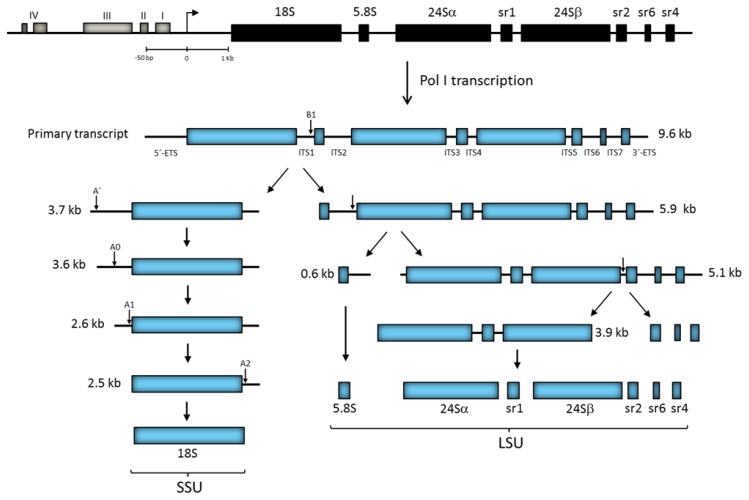
Schematic overview of rRNA transcription and processing in *T. brucei*. The top part represents a rRNA gene repeat, showing the position of the genes encoding the eight rRNA species. The four domains (I to IV) that comprise the promoter region are indicated. The arrow represents the transcription start site. Please note that the scales are different upstream and downstream of the transcription start site. After transcription by Pol I, a primary transcript is generated (9.6 kb). The location of 5′- and 3′-UTRs and ITSs (1 to 7) is shown. Processing of the primary transcript produces the eight mature rRNA species, which are part of the SSU (18S) or the LSU (5.8S, 24Sα, sr1, 24Sβ, sr2, sr4, and sr6). The positions of the main cleavage sites are denoted with small vertical arrows.

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
