# Peer review of "Nucleolar Structure and Function in Trypanosomatid Protozoa"

_cells, 2019, doi:10.3390/cells8050421_

Round 1

Reviewer 1 Report

The review "Nucleolar structure and function in trypanosomatid Protozoa is well written by Martinez-Calvillo and collaborators and provides a very nice overview for the outsider of the nucleolus as well as rRNA processing.

My main comment is relatively minor:

Lines 535-536, authors say that: “In this context, it seems evident that the nucleolus actively contributes to the protection and conservation of essential RBPs and other mRNAs regulators when cells are under dangerous stress.”

Although it is clear that RBP and mRNAs are translocated to nucleolus after stress, I could not see the experiment that show that in the nucleolus these factors are free from degradation. It should be better explored in the text.

In addition, the review could really benefit from a future outlook section: questions that are interesting to be approached, for instance.

Author Response

Reviewer 1

1. Lines 535-536, authors say that: “In this context, it seems evident that the nucleolus actively contributes to the protection and conservation of essential RBPs and other mRNAs regulators when cells are under dangerous stress.” Although it is clear that RBP and mRNAs are translocated to nucleolus after stress, I could not see the experiment that show that in the nucleolus these factors are free from degradation. It should be better explored in the text.

Response: We agree with the reviewer that the results mentioned and discussed in this paragraph do not allow us to firmly conclude that the mRNAs and RBPs are protected from degradation in the nucleolus. Accordingly, we have added more information and rephrased the paragraph in order to avoid overstatements.  The new paragraph is: “Furthermore, in epimastigotes of T. cruzi a set of cytoplasmic mRNAs is translocated into the nucleolus in response to cellular stress generated by severe heat shock exposure [169]. Moreover, while long-term incubation with actinomycin D results in cytoplasmic mRNA decay, it also causes nucleolar mRNA accumulation, which suggests a potential role of the nucleolus in the protection of mRNAs under stress conditions [167]. Thus, as in other eukaryotes, the trypanosomatid nucleolus seems to coordinate different cellular responses to confront the plethora of cellular stressors, indicating that this bipartite nuclear body participates in non-ribosomal functions.”

2. In addition, the review could really benefit from a future outlook section: questions that are interesting to be approached, for instance.

Response: We have included a “Future outlook” section.

Reviewer 2 Report

The manuscript submitted by Martinez-Calvillo et al. is a review on nucleolar structure and function in the parasites from the Trypanosoma family. The review provides a comprehensive overview of functions of the nucleolus, underlining the differences between human (and other vertebrates) and Trypanosomatids. In particular, the manuscript goes over core functions of the nucleolus, ie the structure of the nucleolus, rRNA transcription, regulation and processing, as well as ribosome biogenesis. There is a small section on other functions of the nucleolus such as stress response and processing of mRNAs encoding procyclins, but not as much is known on these functions in Trypanosomes.

Overall the review is well written, very thorough and easy to follow. The literature cited is accurate and extensive. The three figures are appropriate and of high quality.

Overall, I recommend this manuscript for publication.

Author Response

Reviewer 2 did not express any concerns.